# Validation of Automated Countermovement Vertical Jump Analysis: Markerless Pose Estimation vs. 3D Marker-Based Motion Capture System

**DOI:** 10.3390/s24206624

**Published:** 2024-10-14

**Authors:** Jelena Aleksic, Dmitry Kanevsky, David Mesaroš, Olivera M. Knezevic, Dimitrije Cabarkapa, Branislav Bozovic, Dragan M. Mirkov

**Affiliations:** 1Faculty of Sport and Physical Education, University of Belgrade, 11000 Belgrade, Serbia; jelena.aleksic@fsfv.bg.ac.rs (J.A.); olivera.knezevic@fsfv.bg.ac.rs (O.M.K.);; 2Newstream, 3 Rapaport Street, Kfar Sava 4465141, Israel; dima@annoviz.com; 3School of Electrical Engineering, University of Belgrade, 11000 Belgrade, Serbia; davidmesaros9905@gmail.com; 4Jayhawk Athletic Performance Laboratory—Wu Tsai Human Performance Alliance, Department of Health, Sport and Exercise Sciences, University of Kansas, Lawrence, KS 66045, USA; dcabarkapa@ku.edu

**Keywords:** MMPose, jump, motion tracking, accuracy, center of mass, biomechanics

## Abstract

This study aimed to validate the automated temporal analysis of countermovement vertical jump (CMJ) using MMPose, a markerless pose estimation framework, by comparing it with the gold-standard 3D marker-based motion capture system. Twelve participants performed five CMJ trials, which were simultaneously recorded using the marker-based system and two smartphone cameras capturing both sides of the body. Key kinematic points, including center of mass (CoM) and toe trajectories, were analyzed to determine jump phases and temporal variables. The agreement between methods was assessed using Bland–Altman analysis, root mean square error (RMSE), and Pearson’s correlation coefficient (r), while consistency was evaluated via intraclass correlation coefficient (ICC 3,1) and two-way repeated-measures ANOVA. Cohen’s effect size (d) quantified the practical significance of differences. Results showed strong agreement (r > 0.98) with minimal bias and narrow limits of agreement for most variables. The markerless system slightly overestimated jump height and CoM vertical velocity, but ICC values (ICC > 0.91) confirmed strong reliability. Cohen’s d values were near zero, indicating trivial differences, and no variability due to recording side was observed. Overall, MMPose proved to be a reliable alternative for in-field CMJ analysis, supporting its broader application in sports and rehabilitation settings.

## 1. Introduction

Countermovement vertical jump (CMJ) is a widely used assessment modality in various fields of human movement science, including sports [1] and rehabilitation [2]. Analyzing the biomechanical aspects of vertical jumps can provide valuable insights into an athlete’s explosive strength and lower-limb mechanical capacities [3,4]. Furthermore, this analysis can reveal imbalances or asymmetries in lower-limb function, which is crucial for identifying functional performance deficits and understanding injury-risk mechanisms [5]. Traditionally, the gold standard for vertical jump biomechanical analyses has involved the use of marker-based motion capture (MoCap) systems [6] and force plates [7]. These methods offer high precision but are often limited by their high cost, complexity, and requirement for specialized laboratory settings and trained personnel [6]. Consequently, there has been a growing interest in developing more accessible solutions that can provide comparable accuracy while being easier to use in field-based environments [8,9].

Advancements in computer vision and machine learning have made it possible to estimate human poses from smartphone video recordings without the need for placing reflective markers on the subject [10,11]. Consequently, the use of markerless pose-estimation tools has garnered significant attention from sports science researchers in recent years. Particularly in the context of vertical jump assessments, the MyJump app has been the subject of multiple validation studies, demonstrating high levels of correlation (r > 0.95) with traditional gold-standard methods (i.e., force plates) for measuring jump height [11,12,13,14,15]. However, studies also noted that this app tends to overestimate jump height by an average of 4.32 cm compared to the gold-standard testing modalities [15], and unlike force plates, its major limitation pertains to the small number of kinetic and kinematic performance parameters that it can provide [16].

Conversely, tools like OpenPose [17], BlazePose [18], OpenCap [10], and MMPose [8] leverage deep learning-based algorithms to offer advanced capabilities for biomechanical analyses. These tools enable simultaneous tracking of multiple joints and extraction of a wide range of biomechanical variables essential for in-depth performance assessments [19,20]. For instance, analyzing joint angles at the ankle, knee, and hip during the take-off and landing phases of a jump can reveal asymmetries or weaknesses that may predispose an athlete to a greater risk of lower-limb injury [21]. Additionally, variables related to jump phase durations and jump height provide valuable insights into an athlete’s explosive power and neuromuscular efficiency, which are crucial for optimizing training and tracking athletes’ progress over time [4,22].

Validating markerless pose estimation frameworks could address several practical challenges in sports biomechanics, such as enabling comprehensive kinematic evaluations directly on the field and in real time [20,23]. These tools could also be seamlessly integrated into common field tests like horizontal or single-leg jumps, which are often used to evaluate lower limb function and return-to-sport readiness after anterior cruciate ligament (ACL) reconstruction [2]. Such developments would not only improve access to high-quality data but also facilitate timely and individualized interventions that could enhance performance and mitigate injury risk. To date, OpenPose has been the most extensively researched pose-estimation framework in sports biomechanics and validated for tasks such as walking [24,25], running [26,27], cycling [28], and rowing [29]. The average detection accuracy of this framework has been reported to be 94.5% [20]. Conversely, in a study comparing different pose estimation frameworks, MMPose proved to be the most effective model for tracking lower-limb kinematics during a walking task (r = 0.76–0.99; RMSE = 4.5–18.1 mm) [8]. Regarding vertical jump task, several studies have demonstrated good reliability and validity of tools like OpenCap and OpenPose in tracking lower limb kinematics (r = 0.51–0.94; RMSE = 11.6°–14.7°) [10,30], as well as ground reaction forces (MAE = 6.2%, 5–15% bias) [9,31] and joint torques (MAE = 1.2%) [9].

Despite the promising results from existing scientific literature, the majority of the studies has been focused on studying less dynamic, locomotion tasks [8,20,24,25,27,28,29]. Additionally, many studies utilizing markerless pose estimation methods have been limited by small sample sizes [32,33] or relied on remotely captured video data [20,25]. Research on vertical jumps has primarily focused on OpenCap and OpenPose models, which demand significant computational resources [8], while often assessing a narrow range of variables [9,10,26,30,31], thereby limiting their practicality and applicability in real-world sports contexts.

This research introduces a method for the automated temporal analysis of the countermovement vertical jump utilizing the MMPose pose estimation framework [34], with RTMPose 2D top-down pose estimation model. Specifically, the aim of the present study was to assess and confirm the validity of this automated method compared to a gold-standard 3D motion capture system. Instead of identifying specific jump events based on joint locations, this observation was limited to the 2D center of mass (CoM) movement and vertical displacement of the toe marker to identify specific jump phases and associated variables. The findings of this study could pave the way for broader applications of markerless pose estimation in human motion science, ultimately enhancing our understanding of athletic movement and optimizing sports performance analysis and rehabilitation.

## 2. Materials and Methods

### 2.1. Experimental Session

Twelve healthy, physically active, participants ($\underline{x}$ ± SD: age = 25.6 ± 3.4 years; height = 179.1 ± 8.3 cm; body mass = 76.0 ± 16.0 kg) volunteered to participate in the present study. The study included 7 male ($\underline{x}$ ± SD: age = 25.3 ± 3.5 years; height = 183.6 ± 4.7 cm; body mass = 80.8 ± 9.4 kg) and 5 female participants ($\underline{x}$ ± SD: age = 26.4 ± 3.7 years; height = 171.7 ± 7.6 cm; body mass = 68.2 ± 3.6 kg). The participants were familiar with the correct CMJ technique prior to this research. All participants provided written informed consent in accordance with the guidelines of the University’s Institutional Ethical Review Board (Approval #02-848/23-2; Date: 5 May 2023) and the Declaration of Helsinki. The required sample size was determined using G*Power (ver. 3.1.9.7; Heinrich-Heine-Universität Düsseldorf, Düsseldorf, Germany) software. The required sample size for statistical power (1-β) of 0.8, significance level (α) of 0.05, and an effect size of 0.8 was 9 participants.

The measurement setup is shown in Figure 1. Both marker-based motion capture and smartphone camera systems were utilized simultaneously, including 5 Miqus M3 cameras (Qualisys Medical AB, Gothenburg, Sweden) recording at 300 Hz, and 2 iPhone cameras (iPhone 13, Apple Inc., Cupertino, CA, USA) recording at a frame rate of 240 fps (1080p HD). The Qualisys Miqus M3 cameras were positioned around the subject at a 4 × 4 m radius to capture full-body motion. The iPhone cameras were positioned 2 m away from the participants on each side (right and left) and mounted on a tripod set up at a height of 1.10 m. This tripod height was chosen as the most suitable height to align closely with the participants’ central body part (abdomen), ensuring the phone was perpendicular to the subject and thus minimizing the influence of perspective on pose estimation accuracy.

The marker-based system was calibrated to ensure that the origin of the X–Z coordinate system approximately aligns with the point on the surface where the center of mass (CoM) is vertically projected. The X-axis was oriented in the posterior–anterior direction, and the Z-axis was directed upward. After calibration, the origin was marked on the floor, and the subjects always took the same position, with their feet aligned with this point. The markerless model was always calibrated relative to body height, with the X–Z plane dividing the subject into left and right halves.

The subjects were instructed to wear minimal, tightly fitting, dark-colored clothing to create a better contrast with the background, as well as facilitate the placement of the reflective markers. Lightweight infrared reflective markers (i.e., 19 mm diameter) were placed bilaterally on 14 anatomical landmarks (Figure 1). These landmarks included the shoulder (acromion), hip (greater trochanter), knee (lateral condyle of the femur), ankle (lateral malleolus), heel, small toe, and big toe. This marker setup ensured accurate tracking of segmental body movements during the countermovement jumps (CMJs).

Participants first completed a standardized warm-up protocol, which included 3 min of cycling and dynamic stretching, followed by 2–3 familiarization jump trials before commencing experimental trials. The participants were instructed to perform the CMJ exercise with their hands placed on their hips and the correct jump technique was demonstrated and explained to each participant. Subsequently, they completed five jump trials, with a rest period of 10–15 s to minimize the effects of fatigue. Each jump was recorded simultaneously using marker-based motion capture and smartphone camera systems. The beginning of the recording was synchronized manually by using a wireless camera remote shutter and starting the recording at the same time in Qualisys Track Manager software (QTM ver. 2024; Qualisys Medical AB, Gothenburg, Sweden). All sessions were conducted under standardized laboratory conditions for all participants.

### 2.2. Data Processing

#### Motion Capture Recordings

Marker-based motion capture recordings were processed using the QTM software (ver. 2024). A custom kinematic model was built for each participant, consisting of 14 marker positions strategically placed on key anatomical landmarks (i.e., right and left: shoulder, hip, knee, ankle, heel, small toe, and big toe) to accurately capture lower body and joint movements during CMJs. During the processing phase, any missing marker data due to occlusions or tracking errors were corrected using linear or polynomial interpolation in QTM software, ensuring continuity in the motion trajectories. Where necessary, the trajectories were smoothed using a moving average filter, to reduce high-frequency noise and enhance signal clarity without distorting the kinematic data. The processed trajectories were exported as .tsv files for subsequent analysis.

### 2.3. Markerless Data

The workflow for processing smartphone video recordings is outlined in Figure 2. Markerless pose estimation was carried out using the OpenMMLab Pose Estimation Toolbox (MMPose; Version 1.3.2), a deep-learning framework built on PyTorch. The video footage was resampled to 100 Hz to ensure consistent frame rates for analysis. Person detection was then conducted using the RTMDet model, which identifies individuals within each video frame. To maintain accurate tracking and avoid detector misses, especially in sequences where the person might briefly occlude or move rapidly, we integrated a pixel-level object tracker using the MOSSE tracker in OpenCV, paired with deepocsort for ID tracking. This approach ensured that only the main person in the video was consistently tracked throughout the sequence. Subsequently, pose keypoint estimation was carried out using Real-Time Model for Pose Estimation (RTMPose) [23], which accurately localizes key points on the human body and converts them into 2D coordinates. The key point coordinates were then gap-filled and smoothed using an Exponential Moving Average filter to reduce noise and enhance the accuracy of the trajectories. Finally, the refined key point coordinates were exported in a format compatible with Qualisys data (.tsv), allowing for further in-depth kinematic analysis.

### 2.4. Computation of CoM

The positions and velocities of the segment and full-body CoM were quantified frame-by-frame in MATLAB ver. 16 (MathWorks, Natick, MA, USA) for both marker-based and markerless datasets. The overall CoM (i.e., body CoM) was calculated using the Dempster model (1955), which applied segment weights to various body parts: foot (1.5%), lower leg (4.65%), upper leg (10%), and upper body (43%). This model was selected as it provides an accurate representation of the body’s movement dynamics during the vertical jump task and is consistent with the methodologies used in previous studies [35,36]. The use of this model allows us to quantify the kinematics and dynamics of the jump in a comprehensive manner, as it captures the collective contribution of all body segments.

Firstly, the segment centers (X and Z coordinates) were calculated for the foot (CoM was at the midpoint between the heel and toe markers), the lower leg (CoM was at 43.3% of the length from the ankle to the knee marker), the upper leg (CoM was at 43.3% of the length from the knee to the hip marker), and the upper body (CoM was at the midpoint between the hip and shoulder markers). Subsequently, the X and Z coordinates for each segment (i.e., foot, lower leg, upper leg, and upper body) were used to calculate the overall CoM using the following formulas:$$X_{CoM}=\frac{\left(1.5\%\times X_{foot}\right)+\left(4.65\%\times X_{lower\;leg}\right)+\left(10\%\times X_{upper\;leg}\right)+\left(43\%\times X_{upper\;body}\right)}{1.5\%+4.65\%+10\%+43\%}$$
$$Z_{CoM}=\frac{\left(1.5\%\times Z_{foot}\right)+\left(4.65\%\times Z_{lower\;leg}\right)+\left(10\%\times Z_{upper\;leg}\right)+(43\%\times Z_{upper\;body})}{1.5\%+4.65\%+10\%+43\%}$$

### 2.5. Variables

The choice of variables for this study was informed by prior research [7,12] and included temporal variables pertaining to the duration of the specific phases of the jump (Table 1): the unweighing phase (s), braking phase (s), eccentric phase (s), propulsive phase (s), take-off phase (s), landing eccentric phase (s), and flight time (obtained from the toe marker vertical displacement/time trajectory; [s]). Other variables included the countermovement depth (m), landing depth (m), max/min CoM vertical velocity (m/s), take-off CoM vertical velocity (m/s), ankle/knee/hip flexion angles (deg), and jump height (m). The smallest joint flexion angles at the ankle, knee, and hip were measured when transitioning from the eccentric to the concentric phase both during the push-off and landing phases of the jump. For further details regarding the temporal variables, refer to Table 1.

For both the marker-based and markerless datasets, key points on the CoM and the big toe (Toe) trajectories were tracked to identify specific jump phases and extract relevant temporal variables, focusing particularly on the Z-t (vertical displacement–time) and Vz-t (vertical velocity–time) curves (Figure 3). The toe marker was crucial for determining when the feet left the ground and when ground contact was reestablished.

As illustrated in Figure 3, key phases of the jump are marked by 9 specific points: At point a, the CoM begins downward movement, which is identified when CoM velocity exceeds 5% of the maximum velocity. The velocity then reaches its first minimum at point c, corresponding to a transition phase before the upward thrust. Point d indicates the lowest CoM position (first minimum of CoM height) before take-off, followed by reaching the maximum CoM velocity at point e. Then, the toe marker lifts off the ground at point f, signifying the start of the flight phase of the jump. The maximum CoM height occurs at point g. As the jump concludes, the CoM and toe descend, returning to their initial positions at point h, with the CoM reaching its second lowest point at i (second minimum of CoM height).

### 2.6. Statistical Analysis

The similarity between the results of the marker-based and markerless methods was evaluated using Bland–Altman bias and limits of agreement (LoA) [37]. Additionally, the root mean square error (RMSE) and Pearson’s correlation coefficient (r) were calculated. The correlation (r) thresholds were defined as follows: r < 0.3 (negligible), r = 0.3–0.5 (low), 0.5–0.7 (moderate), 0.7–0.9 (high), and 0.9–1.0 (very high correlation) [38]. Bias, LoA, and r were computed using a MATLAB toolbox [39]. The variables analyzed included the center of mass (CoM) Z-t, Toe Z-t, ankle angle–time, knee angle–time, and hip angle–time. The consistency of all CMJ temporal variables obtained from both methods was further assessed using the intraclass correlation coefficient (ICC 3,1). Potential systematic differences were evaluated using a two-way repeated-measures ANOVA with factors: method (marker-based vs. markerless) and side (left vs. right). Cohen’s effect size (d) was applied to quantify the practical significance of the differences, with thresholds defined as follows: d < 0.2 (trivial or no effect), d = 0.2–0.5 (small), d = 0.5–0.8 (moderate), d = 0.8–1.3 (large), and d > 1.3 (very large) [40]. Cohen’s d effect size (ES) and 95% confidence interval (CI) were computed using the harmonic mean of the standard deviations (SDs) of the compared conditions, with an ES of 0.20 considered the minimal value of practical importance. When the 95% CI overlapped with substantial positive and negative values, the effect was deemed unclear; otherwise, the effects were deemed clear. Statistical significance was set at *p* < 0.05. Magnitude comparison analysis was performed using a custom Excel spreadsheet (available from https://www.sportsci.org/jour/03/wghtrials.htm, accessed on 18 September 2023), while other statistical analyses were conducted using SPSS software (IBM SPSS Version 20.0, Chicago, IL, USA).

## 3. Results

The comparison between the marker-based and markerless methods revealed high levels of agreement across all analyzed variables, as presented in Table 2 and illustrated in Figure 4a,b. For the Toe Z-t variable, the bias was minimal, with values of −0.021 m for the left side and −0.020 m for the right side. The LoA ranged from −0.073 m to 0.032 m for the left side and −0.070 m to 0.030 m for the right side. The RMSE values were 0.035 m for the left side and 0.034 m for the right side, with a very high correlation coefficient (r) of 0.992 and 0.993 for the left and right sides, respectively.

For the CoM Z-t variable, the bias was nearly zero, with 0.000 m for the left side and −0.001 m for the right side. The LoA ranged from −0.035 m to 0.035 m for the left side and −0.040 m to 0.037 m for the right side. The RMSE values were 0.021 m and 0.022 m for the left and right sides, respectively, with an exceptionally high correlation coefficient (r) of 0.999 for both sides.

The ankle angle–time (Angle-t) analysis showed a slight bias, with −0.49° for the left side and −0.67° for the right side. The LoA ranged from −8.6° to 7.7° for the left side and −9.5° to 8.143° for the right side. The RMSE values were 5.4° for the left side and 6.487° for the right side, with very high correlation coefficients (r) of 0.984 and 0.981, respectively.

For the knee angle–time variable, the left side showed a bias of 2.6° with LoA from −6.7° to 11.9°, while the right side showed a bias of 2.01° with LoA from −7.3° to 11.309°. The RMSE values were 6.9° and 7.163°, with very high correlation coefficients (r) of 0.994 for both sides.

Lastly, for the hip angle–time variable, the left side showed a bias of 3.9° with LoA from −6.1° to 14.024°, and the right side showed a bias of 5.6° with LoA from −3.9° to 15°. The RMSE values were 8.0° for the left side and 8.8° for the right side, with very high correlation coefficients (r) of 0.997 and 0.996, respectively.

Table 3 presents the results for the temporal CoM variables. The within-within ANOVA revealed no significant differences between the sides (left vs. right) across the temporal phases analyzed, indicating that the side from which the recording was made did not influence the results. The unweighting phase duration showed no significant difference between the marker-based and markerless systems, with ICC values of 0.912 (left) and 0.914 (right), indicating excellent reliability. The breaking phase, eccentric phase, and propulsive phase showed statistically significant differences between the two methods, with the propulsive phase exhibiting the largest F-value (659.964), indicating a strong difference in duration between the methods. Despite these differences, ICC values remained high across most phases, demonstrating strong agreement between methods.

For jump height, the markerless system slightly overestimated the height compared to the marker-based system, with significant differences observed (F_1,110_ = 90.965, *p* < 0.01). Nevertheless, the ICC values were high, indicating that the markerless method still provides reliable measurements. Similarly, for the maximum and minimum CoM vertical velocity, statistically significant differences were found between the two methods, but the ICC values confirmed high reliability.

Cohen’s d analysis, as illustrated in Figure 5, further supports these findings by quantifying the magnitude of differences between the marker-based and markerless methods. Most variables show Cohen’s d values close to zero, indicating that the differences between the methods are not practically relevant. This suggests that the markerless method produces results highly comparable to those from the marker-based method. Importantly, the analysis shows that the side from which the recording was made—whether left or right—did not introduce significant variability, as Cohen’s d values for both sides are similar and generally small.

For example, Cohen’s d for the unweighting phase was approximately 0.001, indicating no meaningful difference between the methods. The breaking phase had small effect sizes, with Cohen’s d values around −0.15 to −0.18, suggesting slight but not practically relevant differences. The propulsive phase exhibited a moderate effect size (d ~ −0.80), which indicates a more noticeable difference between the methods for this specific phase. However, this difference is still within a range where the overall agreement between the methods remains strong, supported by the high ICC values.

Moreover, variables such as jump height and flight time exhibited small-to-moderate Cohen’s d values, reinforcing that while there may be some differences in these specific measures, they are not large enough to be considered practically significant in most contexts. The high correlation coefficients and small effect sizes across most variables emphasize that the markerless approach is robust and consistent with the marker-based system, regardless of the side from which the data were recorded.

## 4. Discussion

This study aimed to evaluate the validity of using MMPose, a markerless pose estimation model, for automated temporal analysis of CMJs by comparing it against the 3D motion capture system (i.e., Qualisys). The comparison between the two methods of assessment demonstrated a high level of agreement across all analyzed biomechanical variables, supporting the validity of MMPose for analyzing CMJ. Specifically, variables related to the CoM movement and joint angles exhibited minimal bias, narrow LoA, and low RMSE, indicating a strong agreement between the two systems. A key finding of this study is the negligible bias and high correlation coefficients observed for the CoM Z-t variable between the two methods. Bias was almost nonexistent for both the left and right sides, with LoA values closely aligned, and RMSE values as low as 0.021 m and 0.022 m for the left and right sides, respectively. The perfect correlation (r = 0.999 for both sides) further confirms the robustness of the markerless method in tracking CoM during the CMJ task. These results align with those of Van Hooren et al. (2023), who also found high accuracy in CoM tracking using markerless systems. Their study highlighted that modern markerless systems can achieve accuracy levels that closely approximate traditional marker-based systems, particularly in straightforward movements like CMJ [26]. Notably, the results suggest that the side from which the 2D CoM was extracted is irrelevant, as both sides produced nearly identical results. This finding is significant because it implies that data from either side can be used interchangeably in future analyses, providing greater flexibility in data collection and analysis. This also simplifies the process, reducing the need to standardize the recording side, which can be particularly advantageous in field settings or when dealing with large datasets.

While the overall agreement between the two methods was found to be strong, slight differences were noted in the Toe Z-t variable and joint angle measurements (i.e., ankle, knee, and hip). For example, the bias for the Toe Z-t variable was minimal (−0.021 m for the left side and −0.020 m for the right side), but the LoA and RMSE values indicated slightly more variability than for the CoM. The ankle, knee, and hip angle–time variables also revealed small biases and RMSE values, with correlation coefficients remaining high (r > 0.98), though slightly lower than those observed for CoM. These differences may be due to the distinct methodologies employed by the marker-based and markerless motion capture systems. In the Qualisys system, physical markers are attached to the body at specific anatomical landmarks, and their motion is tracked in a 3D space. In contrast, MMPose uses a learnable algorithm to identify and track key points on the body from a 2D video feed. The slight discrepancies in the motion of the Toe marker and joint angles likely stem from differences in the exact positioning of these markers. The location where a marker is attached in the MoCap system may not perfectly align with the position that MMPose identifies based on visual cues in the video. This observation is supported by findings from several authors [33,41], who noted that markerless systems, while generally reliable, can exhibit small but systematic differences in joint angle measurements due to their reliance on visual recognition rather than physical marker placement. In addition, Boldo et al. (2024) highlighted the potential for learnable models like MMPose to reduce these discrepancies over time as the system is exposed to more diverse datasets and refined through machine learning techniques [41].

One notable aspect of the MMPose system is in its adaptive learning capability [41,42]. As a machine learning-based model, MMPose has the potential to enhance its accuracy with exposure to larger and more diverse datasets. The findings of Cronin (2021) support this perspective by showing that markerless motion capture systems, particularly those enhanced through machine learning, can substantially improve accuracy over time. Their study demonstrated that with iterative training and refinement, these systems could rival the precision of traditional marker-based approaches in specific kinematic analyses [42]. Therefore, if a broader range of body types, movement styles, and environmental conditions were included in the training dataset, MMPose could refine its recognition of key points, potentially reducing the observed discrepancies in marker positioning. This adaptability makes MMPose a promising tool for future research and practical applications, as it can continuously improve and provide more precise analyses with further development. Moreover, future research should focus on elaborating these differences, particularly by examining the variability in marker placement and its impact on motion capture accuracy. Understanding these nuances will be crucial for optimizing the use of MMPose in both research and applied settings, such as sports performance analysis and rehabilitation.

One of the main strengths of the present study lies in the automatic detection of temporal variables from the CoM, Toe, and angle traces extracted from the valid markerless 2D pose detection. This analysis approach of temporal variables offers crucial insights into the biomechanics of CMJ. The eccentric phase, which includes the unweighting and braking phases, is essential for preparing the body for explosive take-off during a CMJ [3,22]. The findings reveal no significant difference between the marker-based and markerless methods of assessment during the unweighting phase of the movement, with excellent reliability indicated by high ICC values (ICC = 0.912–0.914). Although the braking phase showed statistically significant differences, the small effect sizes (d = −0.15 to −0.18) suggest that these differences are not relevant in a practical setting. Also, these findings are consistent with other studies, such as those conducted by Van Hooren et al. (2023) and Ito et al. (2022) [26,43]. While markerless systems may exhibit minor discrepancies in capturing rapid deceleration movements, it has been found that these differences generally do not impact the practical utility of the data for performance analysis [26,43]. The take-off phase, which encompasses both the eccentric and propulsive phases, directly affects the height and velocity of the jump [4]. The study revealed a more significant difference in the propulsive phase, with a moderate effect size magnitude (d= −0.80). Despite this, the overall agreement between the marker-based and markerless motion capture methods remained strong, supported by high ICC values (ICC = 0.941–0.982). Research by Mundt et al. (2023) and Tang et al. (2023) corroborates these findings, noting that while markerless systems can struggle with capturing the rapid dynamics of the propulsive phase, they still provide valuable insights comparable to those from marker-based systems [20,44]. The moderate effect size suggests that while practitioners should be cautious when interpreting results from this phase, the markerless system’s measurements remain largely reliable for practical purposes.

In addition, it should be noted that jump height and flight time are critical outcome metrics in CMJ analysis [7,11]. The markerless system in the present study slightly overestimated jump height compared to the marker-based system (4.8–6.2%), with significant differences observed. However, the small-to-moderate effect size values suggest that these statistically significant differences are not practically significant in most contexts. This observation is further supported by multiple research reports [45,46], which indicate that despite slight overestimations, markerless motion capture systems can provide sufficiently accurate measurements for most practical applications. Furthermore, Balsalobre-Fernández (2023) demonstrated that such discrepancies could be corrected through regression techniques or enhanced learning algorithms, further improving the precision of markerless systems over time [32].

However, this study does have certain limitations. One limitation is that the accuracy of the MMPose framework was evaluated primarily within the context of CMJs, which are not highly complex multi-directional body movements. As observed in some studies [47,48], the performance of the markerless system may differ in more complex or varied movement patterns, such as those involving significant lateral or rotational components. Additionally, the study’s sample size and the diversity of movement styles included in the analysis may limit the generalizability of the findings, thus future research should explore the applicability of markerless systems like MMPose across a broader range of movements and in more diverse populations. Moreover, it would be valuable to investigate how well MMPose performs in real-world settings outside of controlled laboratory environments, where factors such as lighting, background, and clothing may introduce additional challenges for pose detection.

Another area for further study is the refinement of MMPose’s algorithm to enhance its accuracy in detecting joint angles and marker positions. Given that slight discrepancies were observed in joint angle measurements compared to the marker-based system, it would be beneficial to explore whether further training of the system on more extensive and diverse datasets could mitigate these issues. Future studies should also focus on developing and validating corrective models, such as regression techniques, to adjust for any remaining biases in the markerless system’s outputs. This observation can be generalized to other findings; where differences between marker-based and markerless systems exist, they are often correctable. The study by Balsalobre-Fernández (2023) emphasized that applying regression models or improving the system’s learning capabilities with more data can significantly reduce these measurement errors, aligning the results more closely with marker-based systems [32]. This adaptability makes markerless systems like MMPose increasingly viable for real-time, accurate performance analysis.

## 5. Conclusions

This study underscores the strong potential of the markerless MMPose system as a reliable alternative to traditional marker-based methods for analyzing CMJ movement. The findings reveal that MMPose provides highly accurate measurements, particularly for key biomechanical variables such as the CoM and joint angles, which are crucial in understanding biomechanical parameters of jumping motion. The ability of MMPose to automatically detect temporal variables from the CoM, Toe, and angle traces demonstrates its practical utility, making it a viable tool for real-world applications. A major strength of this study is the evidence supporting the use of MMPose in practical settings. The system’s automatic detection capabilities offer reliable data and simplify the motion analysis process, reducing the need for extensive manual intervention or specialized equipment. This efficiency is particularly advantageous in dynamic environments such as sports performance analysis, where quick and accurate feedback is essential. Moreover, the study’s findings suggest that the markerless motion capture system’s robust performance encourages further development and application beyond sports. MMPose has the potential to be adapted for clinical settings, where it could be used for a range of motion assessments, gait analysis, and running tasks. The system’s non-invasive nature and ease of use make it particularly well suited for these purposes, where accurate and efficient motion tracking is critical.

In conclusion, the strong evidence presented in this study suggests that the MMPose markerless system is not only a reliable tool for CMJ analysis but also holds significant promise for broader applications in both sports and clinical settings. As the system continues to evolve, it will likely become an increasingly valuable resource for a wide range of motion analysis tasks, encouraging further innovation and application in these fields.

## Figures and Tables

**Figure 1 sensors-24-06624-f001:**
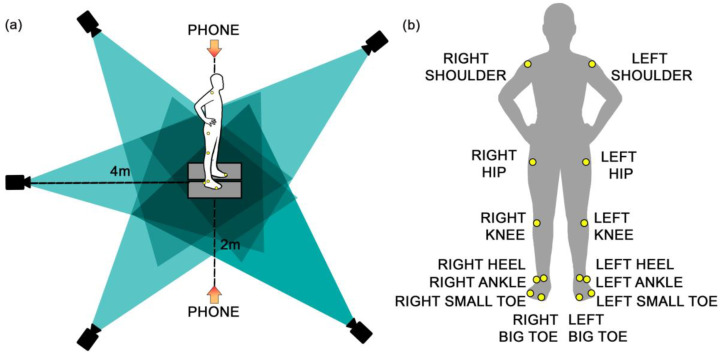
Experimental setup: (**a**) Camera set-up: The black cameras represent a marker-based MoCap system. (**b**) Reflective marker placement: Yellow dots represent the positions of reflective markers on the subject.

**Figure 2 sensors-24-06624-f002:**
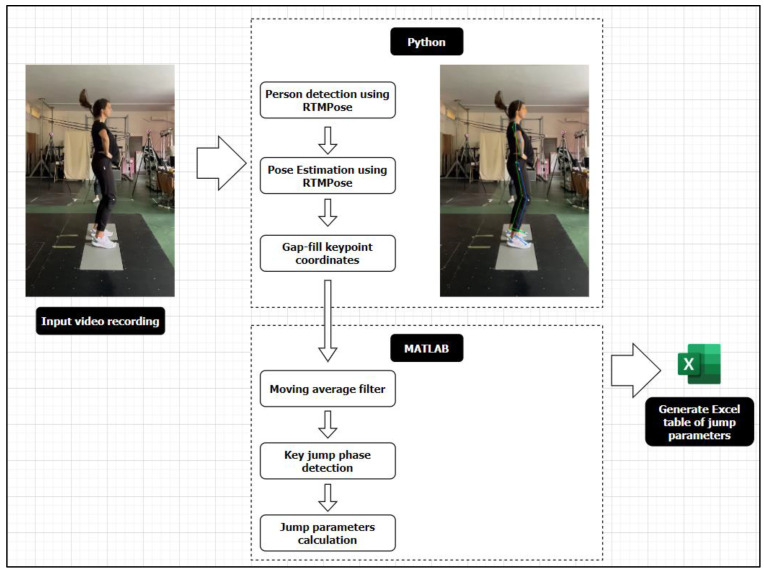
Markerless data processing step-by-step workflow.

**Figure 3 sensors-24-06624-f003:**
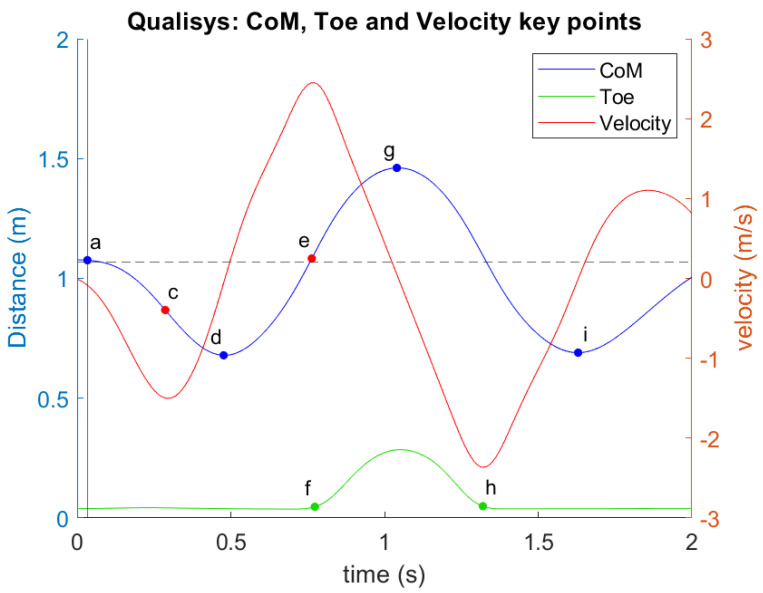
Illustration of Z-t (vertical displacement–time) and Vz-t (vertical velocity–time) curves for Center of Mass (CoM), Toe, and Vertical Velocity During Countermovement Jump. Blue line—CoM, green line—Toe marker height, and red line—Vertical velocity. Key phases of the jump are marked on the curve: (a) Start of downward movement (CoM velocity exceeds 5% of max); (c) The transition before upward thrust (first minimum velocity); (d) Lowest CoM position before take-off; (e) Maximum CoM velocity; (f) Start of flight phase; (g) Maximum CoM height; (h) CoM and toe descend, returning to initial positions; (i) Second lowest CoM position after landing.

**Figure 4 sensors-24-06624-f004:**
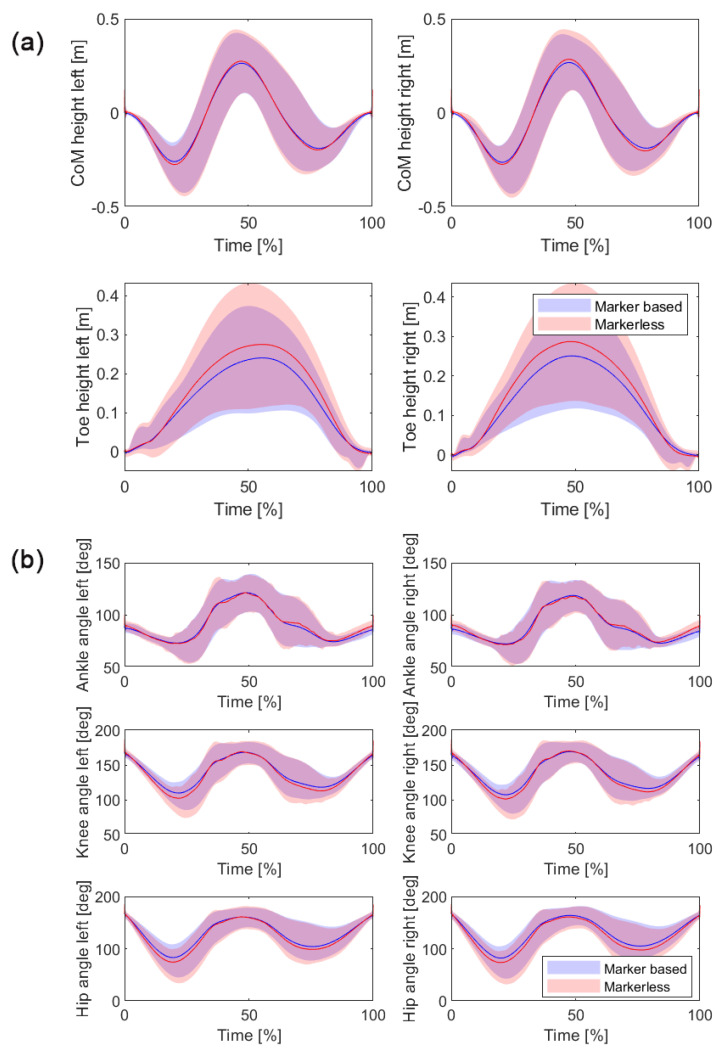
Comparison of time-normalized displacement trajectories between marker-based and markerless systems from both the left and right sides. These trajectories illustrate the movement of (**a**) the center of mass (CoM) and toe height, and (**b**) ankle, knee, and hip angle changes during the entire duration of the countermovement jump (CMJ) task. Solid blue lines represent data from the marker-based system, while red lines represent data from the markerless system. The shaded areas around each line indicate the variability expressed as 95% confidence intervals (CI).

**Figure 5 sensors-24-06624-f005:**
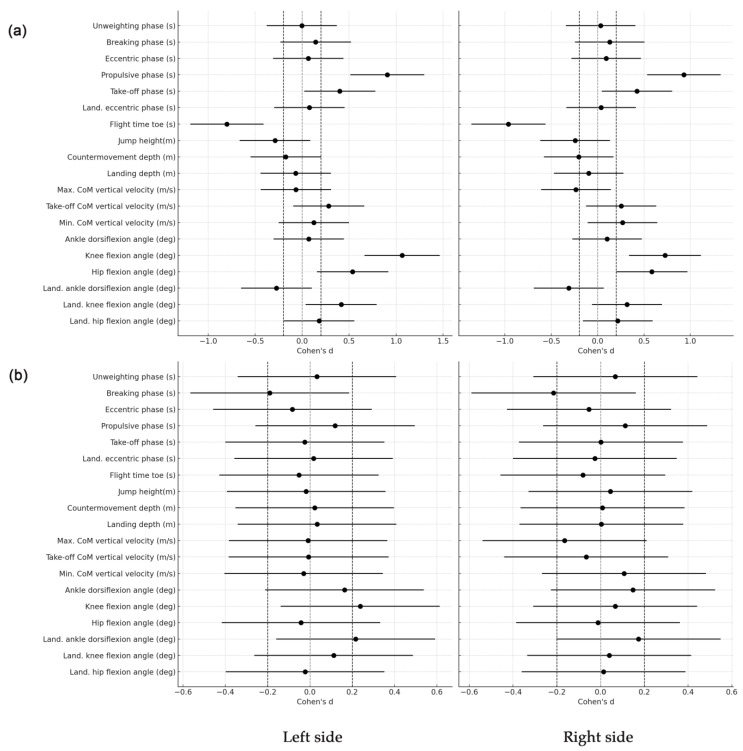
Cohen’s d effect size values and corresponding confidence intervals comparing marker-based (**a**) and markerless (**b**) systems for key temporal CMJ variables obtained from both sides (left and right). The dashed line indicates a small effect size.

**Table 1 sensors-24-06624-t001:** Definitions of countermovement jump variables examined in the present study.

Variable (Unit)	Description
Unweighting phase (s)	The time interval between point a (start of the movement) and point c (start of the braking phase) in the CoM Z-t trajectory.
Breaking phase (s)	The time interval between point c (start of the braking phase) and point d (end of the eccentric phase) in the CoM Z-t trajectory.
Eccentric phase (s)	The time interval between point a (start of the movement) and point d (end of the eccentric phase) in the CoM Z-t trajectory.
Propulsive phase (s)	The time interval between point d (end of the eccentric phase) and point f (take-off) in the CoM Z-t trajectory.
Take-off phase (s)	The time interval between point a (start of the movement) and point f (take-off) in the CoM Z-t trajectory.
LD eccentric phase (s)	The time interval between point h (landing) and point i (end of the landing phase) in the CoM Z-t trajectory.
Flight time (s)	The time interval between point f (take-off) and point h (landing) in the Toe Z-t trajectory.
Jump height (m)	Maximum vertical displacement of the CoM relative to the initial resting position.
Countermovement depth (m)	Minimum vertical displacement of the CoM relative to the initial resting position during the push-off phase.
LD depth (m)	Minimum vertical displacement of the CoM relative to the initial resting position during the landing phase.
Max CoM vertical velocity (m/s)	Maximum vertical velocity of the CoM during the movement.
Take-off CoM vertical velocity (m/s)	Vertical velocity of the CoM at the moment of take-off.
Min CoM vertical velocity (m/s)	Minimum vertical velocity of the CoM during the movement.
Ankle dorsiflexion angle (deg)	The ankle dorsiflexion angle during the transition from the eccentric to the concentric phase of the push-off.
Knee flexion angle (deg)	The knee flexion angle during the transition from the eccentric to the concentric phase of the push-off.
Hip flexion angle (deg)	The hip flexion angle during the transition from the eccentric to the concentric phase of the push-off.
LD ankle dorsiflexion angle (deg)	The ankle dorsiflexion angle during the transition from the eccentric to the concentric landing phase.
LD knee flexion angle (deg)	The knee flexion angle during the transition from the eccentric to the concentric landing phase.
LD hip flexion angle (deg)	The hip flexion angle during the transition from the eccentric to the concentric phase of landing.
Jump height from flight time (m)	Jump height calculated based on the flight time.
Jump height—CoM take-off velocity (m)	Jump height calculated based on the vertical velocity of the CoM at take-off.
Jump height—CoM max. velocity (m)	Jump height calculated based on the maximum vertical velocity of the CoM.

Note: CoM—center of mass; Min—minimum; Max—maximum; LD—landing.

**Table 2 sensors-24-06624-t002:** Comparison between Marker-based and Markerless solutions: Bias, Limits of Agreement, RMSE, and Correlation Coefficients.

		Bias	LoA Lower	LoA Upper	RMSE	Correlation (r)
Toe Z-t (m)	Left	−0.021 (−0.024 ÷ −0.017)	−0.073 (−0.081 ÷ −0.065)	0.032 (0.027 ÷ 0.036)	0.035 (0.031 ÷ 0.038)	0.992 (0.988 ÷ 0.994)
	Right	−0.020 (−0.024 ÷ −0.017)	−0.070 (−0.077 ÷ −0.063)	0.030 (0.026 ÷ 0.034)	0.034 (0.031 ÷ 0.037)	0.993 (0.990 ÷ 0.994)
CoM Z-t (m)	Left	0.000 (−0.004 ÷ 0.003)	−0.035 (−0.041 ÷ −0.030)	0.035 (0.025 ÷ 0.045)	0.021 (0.016 ÷ 0.025)	0.999 (0.998 ÷ 0.999)
	Right	−0.001 (−0.004 ÷ 0.002)	−0.040 (−0.044 ÷ −0.035)	0.037 (0.030 ÷ 0.045)	0.022 (0.019 ÷ 0.025)	0.998 (0.998 ÷ 0.999)
Ankle Angle-t (deg)	Left	−0.49 (−1.45 ÷ 0.5)	−8.6 (−9.7 ÷ −7.6)	7.7 (6.6 ÷ 8.8)	5.4 (5.1 ÷ 5.8)	0.984 (0.982 ÷ 0.986)
	Right	−0.67 (−1.96 ÷ 0.62)	−9.5 (−10.9 ÷ −8.1)	8.143 (6.775 ÷ 9.511)	6.487 (6.019 ÷ 6.956)	0.981 (0.978 ÷ 0.983)
Knee Angle-t (deg)	Left	2.6 (1.4 ÷ 3.8)	−6.7 (−7.9 ÷ −5.5)	11.9 (10.2 ÷ 13.6)	6.9 (6.3 ÷ 7.5)	0.994 (0.993 ÷ 0.995)
	Right	2.01 (0.57 ÷ 3.44)	−7.3 (−8.7 ÷ −5.9)	11.309 (9.440 ÷ 13.178)	7.163 (6.458 ÷ 7.868)	0.994 (0.993 ÷ 0.995)
Hip Angle-t (deg)	Left	3.9 (2.5 ÷ 5.4)	−6.1 (−8.0 ÷ −4.288)	14.024 (12 ÷ 16)	8.0 (6.8 ÷ 9.1)	0.997 (0.996 ÷ 0.997)
	Right	5.6 (3.8 ÷ 7.4)	−3.9 (−5.9 ÷ −1.8)	15 (12 ÷ 17)	8.8 (7.3 ÷ 10.0)	0.996 (0.995 ÷ 0.997)

Note: LoA Lower/Upper–lower/upper limits of agreement; RMSE—root mean square error.

**Table 3 sensors-24-06624-t003:** Descriptive statistics (Mean (SD)) for Temporal Variables obtained by Marker-Based and Markerless solutions from both sides (Left and Right).

	Marker-Based		Markerless		System	Side	Interaction	ICC (95%CI)	
Variable	Left	Right	Left	Right	F(1, 110)	F(1, 110)	F(1, 110)	Left	Right
Unweighting phase (s)	0.241 (0.043)	0.239 (0.044)	0.241 (0.047)	0.238 (0.043)	0.070	0.077	0.115	0.912 (0.851 ÷ 0.949)	0.914 (0.853 ÷ 0.949)
Breaking phase (s)	0.178 (0.032)	0.185 (0.039)	0.173 (0.029)	0.180 (0.032)	5.171 *	1.254	0.003	0.948 (0.912 ÷ 0.970)	0.838 (0.723 ÷ 0.905)
Eccentric phase (s)	0.419 (0.065)	0.424 (0.066)	0.414 (0.067)	0.418 (0.068)	7.537 *	0.131	0.236	0.965 (0.940 ÷ 0.979)	0.988 (0.979 ÷ 0.993)
Propulsive phase (s)	0.243 (0.038)	0.239 (0.039)	0.210 (0.036)	0.206 (0.031)	659.964 **	0.405	0.045	0.982 (0.969 ÷ 0.989)	0.941 (0.900 ÷ 0.966)
Take-off phase (s)	0.662 (0.092)	0.664 (0.096)	0.624 (0.096)	0.625 (0.094)	370.717 **	0.007	0.136	0.985 (0.974 ÷ 0.991)	0.990 (0.983 ÷ 0.994)
LD eccentric phase (s)	0.260 (0.127)	0.258 (0.124)	0.249 (0.135)	0.253 (0.128)	22.443 **	0.001	3.173	0.995 (0.992 ÷ 0.997)	0.996 (0.994 ÷ 0.998)
Flight time toe (s)	0.547 (0.065)	0.553 (0.059)	0.608 (0.086)	0.616 (0.070)	351.538 **	0.292	0.091	0.944 (0.904 ÷ 0.968)	0.926 (0.873 ÷ 0.957)
Jump height (m)	0.340 (0.076)	0.342 (0.068)	0.362 (0.074)	0.359 (0.073)	90.965 **	0.005	1.280	0.971 (0.950 ÷ 0.983)	0.986 (0.975 ÷ 0.992)
Countermovement depth (m)	0.292 (0.079)	0.291 (0.065)	0.306 (0.073)	0.305 (0.075)	16.156 **	0.007	0.021	0.923 (0.868 ÷ 0.955)	0.949 (0.913 ÷ 0.970)
LD depth (m)	0.24 (0.13)	0.24 (0.12)	0.25 (0.13)	0.25 (0.14)	27.490 **	0.009	0.804	0.995 (0.992 ÷ 0.997)	0.991 (0.984 ÷ 0.995)
Max. CoM vertical velocity (m/s)	2.33 (0.31)	2.33 (0.27)	2.35 (0.29)	2.39 (0.27)	20.242 **	0.212	5.519 *	0.961 (0.934 ÷ 0.977)	0.979 (0.964 ÷ 0.988)
Take-off CoM vertical velocity (m/s)	2.33 (0.30)	2.33 (0.26)	2.25 (0.27)	2.27 (0.26)	34.395 **	0.057	0.771	0.945 (0.905 ÷ 0.968)	0.950 (0.914 ÷ 0.971)
Min. CoM vertical velocity (m/s)	−2.23 (0.30)	−2.22 (0.28)	−2.27 (0.31)	−2.30 (0.30)	68.549 **	0.047	8.655 *	0.987 (0.977 ÷ 0.992)	0.981 (0.968 ÷ 0.989)
Ankle dorsiflexion angle (deg)	71 (5)	71 (5)	71 (7)	70 (6)	0.892	0.851	0.024	0.812 (0.680 ÷ 0.890)	0.571 (0.269 ÷ 0.749)
Knee flexion angle (deg)	105.2 (8.4)	103.0 (10.4)	96.8 (7.4)	96.3 (7.8)	170.555 **	0.825	2.245	0.846 (0.737 ÷ 0.909)	0.861 (0.763 ÷ 0.918)
Hip flexion angle (deg)	76 (20)	76 (19)	65 (18)	66 (18)	156.399 **	0.022	0.124	0.946 (0.908 ÷ 0.968)	0.934 (0.888 ÷ 0.961)
LD ankle dorsiflexion angle (deg)	71.3 (5.1)	70.1 (5.2)	72.7 (5.7)	71.8 (5.3)	8.200 *	1.490	0.021	0.744 (0.563 ÷ 0.850)	0.376 (−0.064 ÷ 0.634)
LD knee flexion angle (deg)	109 (16)	107 (18)	102 (18)	102 (18)	128.975 **	0.161	1.111	0.970 (0.949 ÷ 0.983)	0.971 (0.950 ÷ 0.983)
LD hip flexion angle (deg)	96 (39)	97 (39)	89 (40)	89 (40)	118.316 **	0.001	0.997	0.992 (0.986 ÷ 0.995)	0.990 (0.982 ÷ 0.994)

Note: * *p* < 0.05; ** *p* < 0.01.

## Data Availability

The data presented in this study are available on request from the corresponding author.

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
