# Peer review of "Validation of Automated Countermovement Vertical Jump Analysis: Markerless Pose Estimation vs. 3D Marker-Based Motion Capture System"

_sensors, 2024, doi:10.3390/s24206624_

Round 1

Reviewer 1 Report

Comments and Suggestions for Authors

The authors must also break down the sample and results for each sex by gender. How many participants were women and how many were men, in order to generalize the results.

Whether participants were strength sports athletes or not, and whether or not they were familiar with the correct CMJ technique. The anthropometric characteristics of the participants also need to be described in greater detail.

The authors should describe in more detail how they selected the sample.

Author Response

Comment 1: The authors must also break down the sample and results for each sex by gender. How many participants were women and how many were men, in order to generalize the results.

Response 1:
Thank you for pointing this out. We recognize the significance of including comprehensive demographic data to improve the generalizability of our results. Accordingly, we have provided a more detailed distribution of participants in the Methods section (page 3, first paragraph, lines 106-109), specifying the number of women and men involved in the study along with the mean and standard deviation values for age, body height, and body mass for each gender.
Considering that the primary goal of this paper is to compare the accuracy between two measurement systems (marker-based and markerless), we have maintained a focus on this comparison within the results section. Although participant performance could potentially vary by gender, our analysis primarily investigates the accuracy of the systems, not the performance outcomes by gender. The markerless model employed is designed to be gender-neutral, suggesting that gender-specific results are not crucial to our study's main objective, which is to assess the comparative accuracy of these systems. This method ensures that our conclusions highlight the systems' reliability and applicability regardless of gender.

Comment 2: Whether participants were strength sports athletes and whether they were familiar with the correct CMJ technique. The anthropometric characteristics of the participants also need to be described in greater detail. 

Response 2: We appreciate your comment and thank you for raising these points. Accordingly, we have updated the Methods section (page 3, first paragraph, lines 104-105, and 108-109) to clarify that the participants were physically active individuals and familiar with the correct CMJ technique prior to participating in this study.
Although it is acknowledged that individual dimensions can influence the accuracy of biomechanical models, the significance of this factor is mitigated in our study by using the same model across both systems. This consistency ensures that any variations in measurements are likely due to differences between participants rather than the model itself. Furthermore, it is common practice in biomechanics to approximate movement using just one point (such as the hip marker) to define the kinematics of motion (in this case, the vertical jump). Therefore, we deemed it unnecessary to measure additional anthropometric dimensions of our subjects. This approach has allowed us to maintain focus on the comparative accuracy of the marker-based and markerless systems, reinforcing the reliability and applicability of our findings.

Comment 3: The authors should describe in more detail how they selected the sample. 

Response 3: Thank you for your valuable suggestion. We appreciate it. We have revised the Methods section (page 3, first paragraph, line 104-109) to include additional information about the selection criteria for participants (ex. healthy, physically active and familiar with the correct CMJ technique). Additionally, we have included the required sample size obtained from the power analysis in G*Power, which further clarifies how the sample size for this study was determined (age 3, first paragraph, line 113-114).

Reviewer 2 Report

Comments and Suggestions for Authors

The manuscript demonstrates that the markerless pose estimation framework, MMPose, can serve as an alternative to 3D marker-based motion capture systems. The entire article supports this conclusion through the collection and calculation of relevant data, making this work quite meaningful. However, there are a few issues that require modification or explanation: 

1. The explanation of the content represented in the two graphs in Figure 4 is not clear enough. Further clarification is needed regarding the meaning of each part and how to conduct a comparative analysis of the data. 

2.  Figure 4 contains some irrelevant content (highlighted by the red box) 

3. In the manuscript, it is stated that there are 14 marker points on the human body, but in Figure 1(b), only 12 marker points are depicted in the schematic.

Author Response

Comment 1: The explanation of the content represented in the two graphs in Figure 4 is not clear enough. Further clarification is needed regarding the meaning of each part and how to conduct a comparative analysis of the data.  

Response 1:
Thank you for your valuable feedback. We appreciate your request for further clarification on the content represented in Figure 4. Accordingly, we have included a more detailed clarification in the description under Figure 4 to help the readers better understand our findings (page 10, lines 293-298). The figure is designed to provide a visual representation of the alignment of trajectories for all subjects, where time is normalized to ensure that all trajectories have the same duration. The solid lines represent the mean position, velocity, or angle over time, while the shaded areas around these lines show the variability, which is represented as 95% confidence intervals.  

Comment 2: Figure 4 contains some irrelevant content (highlighted by the red box). 

Response 2:
Thank you for pointing this out. There was an error introduced by the journal when formatting the manuscript into Sensors template. The template used does not properly align with the original manuscript, and Figure 4b is mixed with Figure 5b in the formatted version. Accordingly, we have corrected this in the revised version of the manuscript (page 10, Figure 4b).

Comment 3: In the manuscript, it is stated that there are 14 marker points on the human body, but in Figure 1(b), only 12 marker points are depicted in the schematic.

Response 3:
We really appreciate you pointing out to this mistake. We have revised Figure 1 to include the two heel markers (left and right) which were not visible in the previous version of the manuscript (page 4, Figure 1b).

Reviewer 3 Report

Comments and Suggestions for Authors

Overall the manuscript is well designed and constructed. There are some minor issues on the methodology. Below are the suggestions to the authors.

1) There is no information on the definition of the coordinate system (origin and axis orientation) for marker-based and maker-less motion capture systems.

2) The calculated “overall CoM” position is not the body CoM and is not generally used. Therefore, please explain why the authors used this point in the analysis of jump motion. 

3) Table 1 may contain incorrect statements. Should the LD eccentric phase be between point h and point i? And also, should the flight phase be between point f and point h?

4) Figure 4b does not show the results of joint angle changes.

Author Response

Comment 1: There is no information on the definition of the coordinate system (origin and axis orientation) for marker-based and marker-less motion capture systems. 

Response 1: 
Thank you very much for your comment. We agree that it is important to provide a clear definition of the coordinate system for both the marker-based and markerless motion capture systems. Accordingly, we have included this information in the Methods section of the revised manuscript for better clarity (page 3, lines 125- 131): "The marker-based system was calibrated to ensure that the origin of the X-Z coordinate system approximately aligns with the point on the surface where the center of mass (CoM) is vertically projected. The X-axis was oriented in the posterior-anterior direction, and the Z-axis was directed upward. After calibration, the origin was marked on the floor, and the subjects always took the same position, with their feet aligned with this point. The markerless model was always calibrated relative to body height, with the X-Z plane dividing the subject into left and right halves."

Comment 2: The calculated “overall CoM” position is not the body CoM and is not generally used. Therefore, please explain why the authors used this point in the analysis of jump motion. 

Response 2: We appreciate your valuable feedback. We understand your concern how the term "overall CoM" can be misinterpreted and have now included additional clarification in the Methods section (page 5, lines 189 - 195). The term "overall CoM" refers to the center of mass calculated for the whole body, based on individual segment contributions. While it may not always be referred to as the "overall CoM" in some literature, this point represents the combined center of mass for the entire body, calculated using a well-established biomechanical model (Dempster, 1955). We chose this particular model for our analysis as it provides a reliable representation of the body's movement dynamics during the jump. This approach is widely used in biomechanical studies, particularly for assessing whole-body motion during activities such as vertical jumps, as it captures the collective contribution of all body segments. Using this model allows us to thoroughly quantify the kinematics and dynamics of the jump, aligning with methodologies from the literature (Petronijevic et al., 2018; Winter et al., 2016), as cited in our revised manuscript. Moreover, as this is a validation study comparing two systems (marker-based and markerless), we believe it is essential to apply the same model to both systems for consistency, which is the approach we have used in this study. 

Comment 3: Table 1 may contain incorrect statements. Should the LD eccentric phase be between point h and point i? And also, should the flight phase be between point f and point h? 

Response 3:
Thank you for pointing this out. We greatly appreciate it. We have updated Table 1 with the correct statements regarding the LD eccentric phase, which is now listed as occurring between points h and i, and the Flight phase, which is now correctly listed as occurring between points f and h (page 6, Table 1). The changes are marked in red.

Comment 4:  Figure 4b does not show the results of joint angle changes.

Response 4:
We appreciate your valuable comment. There was an error introduced by the journal when formatting the manuscript into Sensors template. The template used does not properly align with the original manuscript, therefore the correct graph for Figure 4b, which shows the results of joint angle changes, was not visible in the formatted version. We have revised the manuscript to include the correct graphical representation for Figure 4b (page 10).

Round 2

Reviewer 1 Report

Comments and Suggestions for Authors

The authors have correctly made the suggestions made on my part. Now I think that in my opinion, the work is fine and can be published.